# The Emerging Role of ncRNAs and RNA-Binding Proteins in Mitotic Apparatus Formation

**DOI:** 10.3390/ncrna6010013

**Published:** 2020-03-20

**Authors:** Kei K. Ito, Koki Watanabe, Daiju Kitagawa

**Affiliations:** Department of Physiological Chemistry, Graduate School of Pharmaceutical Science, The University of Tokyo, Bunkyo, Tokyo 113-0033, Japan; ito-delightfully-kei@g.ecc.u-tokyo.ac.jp (K.K.I.); koki.watanabe@mol.f.u-tokyo.ac.jp (K.W.)

**Keywords:** ncRNA, centrosome, kinetochore, mitotic spindle

## Abstract

Mounting experimental evidence shows that non-coding RNAs (ncRNAs) serve a wide variety of biological functions. Recent studies suggest that a part of ncRNAs are critically important for supporting the structure of subcellular architectures. Here, we summarize the current literature demonstrating the role of ncRNAs and RNA-binding proteins in regulating the assembly of mitotic apparatus, especially focusing on centrosomes, kinetochores, and mitotic spindles.

## 1. Introduction

Non-coding RNAs (ncRNAs) are defined as a class of RNA molecules that are transcribed from genomic DNA, but not translated into proteins. They are mainly classified into the following two categories according to their length—small RNA (<200 nt) and long non-coding RNA (lncRNA) (>200 nt). Small RNAs include traditional RNA molecules, such as transfer RNA (tRNA), small nuclear RNA (snRNA), small nucleolar RNA (snoRNA), PIWI-interacting RNA (piRNA), and micro RNA (miRNA), and they have been studied extensively [1]. Research on lncRNA is behind that on small RNA despite that recent transcriptome analysis has revealed that more than 120,000 lncRNAs are generated from the human genome [2,3,4]. LncRNAs can perform diverse functions, including transcription, RNA processing, RNA degradation, and translation [5]. Some lncRNAs function as scaffolds during construction of subcellular structures, such as nuclear bodies [6,7,8,9]. These nuclear bodies are formed by phase separation of RNA-binding proteins with prion-like domain, low complexity region, or intrinsic disordered region [10]. The scaffold ncRNAs of these nuclear bodies, also referred to as architectural RNAs (arcRNAs), bind to and assemble RNA-binding proteins and thereby induce liquid-liquid phase separation [11]. For example, NEAT1 binds to NONO and SFPQ for phase separation of the ribonucleoprotein complex, forming a paraspeckle [12]. These ncRNAs are found widely in eukaryotes, from humans to yeasts [13]. Given the low sequence conservation of these ncRNAs, it is conceivable that the physicochemical properties of ncRNAs are more important characteristics underlying their function for phase separation of the ribonucleoprotein complex.

The fine structure of the mitotic spindle apparatus is critical for cell division and transmission of genetic information. The mitotic apparatus is mainly composed of three structures—centrosomes, kinetochores, and mitotic spindles (microtubules) (Figure 1) [14]. Although the three constituents are basically composed of large numbers of proteins, studies since the 1970s provide evidence that RNA molecules, such as lncRNAs, modulate these structures and their functions [15,16,17,18]. For example, Heidemann et al. (1977) showed that the ability of centrosomes purified from *Chlamydomonas* and *Tetrahymena* to nucleate microtubules in vitro is affected by RNase treatment; this suggests that centrosomes contain RNA molecules that perform essential functions as major microtubule-organizing centers [15]. Three years later, similar results were found in mammalian centrosomes purified from mitotic Ptk2 cells [18]. However, although several studies have revealed the sequence of RNAs localized in the centrosome of the surf clam and mouse [19,20], the critical RNAs responsible for microtubule nucleation in the centrosome have not been identified. The function of ncRNAs transcribed from the centromere region (centromeric RNAs) at the kinetochore has been studied since 2004 and 2007 with maize and human cells, respectively [21,22] Centromeric RNAs are associated with critical kinetochore proteins, recruiting and regulating them at the kinetochore [23]. At present, distinguishing the function of centromeric transcripts and transcription itself in kinetochore formation comes into question [24]. Compared with centrosomes and kinetochores, the function of ncRNAs on the mitotic spindle is poorly understood. Although RNase treatment can disrupt the spindle structure [17], the underlying molecular mechanisms remain unknown. In a study on RNA-binding proteins, Rieder (1979) compared electron microscope images of newt lung cells with and without RNase treatment and revealed that ribonucleoprotein (RNP) can accumulate within kinetochores and centrosomes [25]. Experiments using new technologies, such as immunofluorescence and mass spectrometry, have identified many RNA-binding proteins localized at centrosomes, kinetochores, and mitotic spindles [26].

This review focuses on recent literature demonstrating the functional implications of ncRNA molecules and RBPs in regulating centrosomes, kinetochores, and mitotic spindles.

## 2. Centrosome

### 2.1. Centriole and Pericentriolar Materials (PCM)

The centrosome, which comprises centrioles and PCM, is a non-membrane-bound organelle. Centrosomes serve as the major microtubule-organizing centers (MTOCs) in most animal cells. Therefore, centrosomes contribute to diverse biological processes, including cell division and cell polarity [27].

Centrioles are small barrel-shaped structures characterized by the nine-fold radial symmetry of triplet-microtubules. A mature centriole is ~250 nm in diameter and ~450 nm in height in human cells [28]. The centriole duplication cycle is tightly regulated and coupled with cell cycle progression to ensure the correct number of centrioles and robust formation of the mitotic bipolar spindle [28,29,30]. The centriole structure assembly is an evolutionarily conserved process and is performed by the following five major conserved components—Cep192, Plk4, HsSAS-6, STIL, and CPAP [31,32,33]. During mitosis, centrosomes act as MTOCs to ensure robust formation of mitotic bipolar spindles and proper chromosome segregation. At this stage, the surrounding PCM including PCNT and γ-tubulin drastically expands and acquires MTOC activity [34].

### 2.2. RNAs and RNA-Binding Proteins in Centrosomes

This chapter describes centrosomal RNA and RNA-binding proteins (Figure 2, Table 1). The presence and function of RNAs in the centriole were first noted in the 1970s [25]. To examine the MTOC activity of the centrosome, an aster formation assay using *Xenopus* egg extract was conducted in vitro. Centrioles (basal bodies) purified from *Chlamydomonas* or *Tetrahymena* were treated with several enzymes and subsequently added into the *Xenopus* egg extract. As a result, treatment with protease, RNase A, T1, or S1 nuclease significantly attenuated aster formation activity of centrioles. While protease disrupted the centriole, RNase treatment had no effect on its structure, suggesting that RNAs were not critical components of the centriole structure itself, but were somehow required for the MTOC activity of centrioles. Regarding PCM, RNase was reported to decrease the amount of PCM in mammalian cells [18]. However, the susceptibility of PCM to RNase treatment has been contested in recent years [35].

Recent technological advances have made it possible to identify RNAs localized in the centrosome. In 2006, Alliegro et al. cloned five RNAs enriched in the centrosome of surf clam (*Spisula*) oocytes and named them centrosomal RNA (cnRNA) [19]. Since no matches were found for any of the five cnRNAs in the database by BLAST analysis, these cnRNAs were considered unique sequences. The authors focused on one cnRNA, cnRNA11, which consisted of 3863 nucleotides and was detected in the centrosome by in situ hybridization. Although the anti-sense strand of cnRNA11 possessed an ORF encoding a predicted reverse transcriptase domain, no ORFs were found in the sense strand [19]. A subsequent study identified 120 additional cnRNAs, sixteen of which were related to nucleic acid metabolism or genome structure, such as nucleotide polymerases, microsatellite RNAs, and retroelements. The authors reported that cnRNA65, one of these sixteen cnRNAs, resided in a structure called the nucleolinus before NEBD (nuclear envelope breakdown) and is relocalized to meiotic centrosomes [36]. The internally transcribed spacer region of rRNA is also localized in the nucleolinus and maternal pro-centrosomes of surf clam oocytes [37]. Nucleolinus is an RNA-rich dynamic compartment in the nucleolus that has been poorly studied [38]. This same research group revealed that RNAs associated with the nucleolinus migrate to spindles and centrosomes during cell division and that removing the nucleolinus by laser microsurgery led to failed mitosis and meiosis [39]. In summary, these studies suggest that the nucleolinus and centrosome, key organelles for cell division, share the same RNAs in *Spisula*, but whether these RNAs are required for cell division or organelle organization in this and other species remains unclear.

Regarding vertebrates, genomic-instability-inducing RNA (Ginir), a mammalian lncRNA conserved in rodents potentially involved in centrosome function, was identified as an oncogene in mouse melanoma cells [20]. The Ginir locus is also transcribed as a completely overlapping antisense to Ginir (Giniras). This sense and anti-sense transcript pair is expressed in a spatiotemporal manner during embryonic development and in adult tissues. The authors revealed that Ginir expression is more tightly coupled to proliferative stages of cells during embryonic development, whereas Giniras expression is associated with non-proliferative cells in organs. Mechanistically, the oncogenic function of Ginir is mediated by interaction with centrosomal protein 112 (Cep112). Cep112 interacts with breast cancer type 1 susceptibility protein (Brca1), another centrosome-associated protein. The interaction of the Cep112 protein and Brca1 protein is impaired when cells show elevated levels of Ginir RNA, resulting in severe dysregulation and abnormality in mitosis, leading to malignant transformation [20]. In contrast to Ginir, Giniras does not function as a dominant oncogene in mouse cells but rather when expressed in combination with Ginir, it suppresses its oncogenic function.

In addition to RNAs, a large number of RNA-binding proteins reside in the centrosome. These proteins, such as EWS and NDH2, can be found in a database of protein localization (MicroKits, http://microkit.biocuckoo.org/) [26]. Here, we introduce RBM14, Gle1, HuR, RBM8A, and MAGOH, the functions of which in the centrosome have been recently described. Although these proteins are likely associated with mRNA, it is possible that they also bind to ncRNA to regulate centrosome biogenesis or their associated mRNAs perform a dual role as functional RNA molecules [40,41].

RNA-binding motif 14 (RBM14) was identified as a novel regulator for centriole biogenesis [42]. RBM14 prevents complex formation from two centriole duplication factors, STIL and CPAP, in the cytoplasm through direct interaction with STIL. Depleting RBM14 induces excessive assembly of the STIL-CPAP complex, which acts as a seed to form ectopic aggregates of centriolar proteins in the cytoplasm. Interestingly, this de novo assembly of centriolar protein complexes occurs even in the presence of pre-existing centrioles. Moreover, a portion of the ectopic centriolar protein complexes in turn assemble into structures more akin to centrioles and cause multipolar spindle formation. In contrast, de novo formation of centrioles can be induced under natural or physical loss of pre-existing centrioles. Therefore, RBM14 suppresses the formation of aberrant structures related to centrioles and thereby maintains the integrity of mitotic spindles.

Gle1 regulates DEAD box RNA helicase and is responsible for mRNA export. Depleting Gle1 generates ectopic MTOCs in the cytoplasm, resulting in chromosome mis-segregation in human cells [43]. Gle1 is localized in the PCM, and its depletion significantly attenuates signal intensity of PCNT, a centrosomal scaffold protein, at the centrosome [44]. The mRNA of PCNT has been reported to accumulate in centrosomes awaiting translation [45], but whether Gle1 regulates PCNT mRNA to be translated locally at the centrosome remains to be clarified.

HuR, an RNA-binding protein, may contribute to stabilizing mRNAs by binding to an ARE (AU-rich element) sequence in the 3’UTR region of the target mRNA. In U251 cells, HuR is transferred to the centrosome only when cells are stimulated by growth factors [46]. Simultaneously, the cell undergoes centrosome amplification in the presence of growth factors. Consistent with these observations, the authors showed that the association of HuR with centrosomes and centrosome amplification is positively correlated [46]. The authors also indicated that phosphorylation of HuR by Cdk5 weakens its binding affinity to target mRNAs. Interestingly, phosphorylation-mimicking mutants reduce centrosome numbers, whereas alanine mutants induce centrosome amplification [47].

The heterodimer of RBM8A and MAGOH is a part of exon junction complexes formed on target mRNA. Intriguingly, this heterodimer is localized in the centrosome [48]; depleting either protein led to failed mitotic progression, presumably because of increased frequency of monopolar or multipolar spindle formation [49]. However, whether such mitotic defects stemmed from compromised centrosome function or pre-mRNA splicing remains unknown. Although recent proteomic analyses suggest more RNA-binding proteins exist around the centrosome fraction, identifying their detailed function in centrosome biogenesis and mitosis requires further research.

Recent work suggests that the pericentriolar matrix (PCM) is a phase-separated condensate [50]. SPD-5, a PCM scaffold protein in *C. elegans*, undergoes condensation in vitro and in vivo [51]. The SPD-5 condensates first behave like liquids and rapidly solidify [35,51]. PCM is considered a solid matrix since it resists RNase, salt extraction, and dilution [35,52]. Therefore, PCM probably arises from a phase-separated condensate and matures into a more solid structure. ncRNAs may be involved in PCM nucleation by acting as a seed for phase separation.

## 3. Kinetochore

### 3.1. Components and Regulators of Kinetochores

The kinetochore is the macromolecular protein complex associated with sister chromatids and links them to the microtubule plus-end of mitotic spindles [53]. According to an electron microscopy image, the kinetochore consists of an inner layer, outer layer, and a fibrous corona in vertebrates [54].

Centromeres, the DNA sequences where kinetochore components assemble [55], are embedded in the heterochromatin structure and do not encode any proteins [55]. Each human chromosome has only one centromere, the length of which ranges from several hundred kilobases to several megabases [56]. In the human centromere, there is a centromere-specific sequence of the 171 base pair tandem repeats called alpha-satellite DNA [57]. Although the sequence of centromeric DNA is poorly conserved, the repetitive sequences within the centromere are a common feature among various species, including vertebrates, insects, yeasts, and plants [56]. In addition to the DNA sequence, CENP-A, a centromere-specific histone, is well conserved. CENP-A is a histone H3 variant that forms an octameric complex with histone H2A, H2B, and H4 in the centromere [56]. In human cells, CENP-A is incorporated into the centromere by HJURP, a chaperone protein, within the early G1 phase and is required for assembling inner kinetochore components [53,58].

The constitutive inner-kinetochore proteins are called the “constitutive centromere-associated network” (CCAN) [53]. CCAN proteins reside in the inner kinetochore throughout the cell cycle [59]. CENP-C, one of CCAN proteins, directly binds to CENP-A and centromeric DNA and provides a scaffold for other CCAN proteins [59]. The outer layer of the kinetochore is attached with spindle microtubules termed kinetochore microtubules (k-MTs) and generate the force needed to pull chromatids toward spindle poles by depolymerizing microtubules [59,60].

When the connections between the kinetochore and the spindle microtubules are incorrect, aurora-kinase B weakens the connections by phosphorylating some proteins in the outer layer [61,62]. Aurora-kinase B forms a chromosomal passenger complex (CPC) with INCENP, survivin, and borealin [63]. The CPC is in the inner centromere during early mitosis when the mitotic microtubules are properly attached to the kinetochore [63].

The surveillance mechanism of the kinetochore-microtubule connections is called the “spindle assembly checkpoint” (SAC) [64]. The improper connection assembles SAC-related proteins to the kinetochore and activates the SAC to ensure correct bi-orientation of sister chromatids [64]. Accumulated SAC-related components repress the APC/C-cdc20 ubiquitin ligase complex [64]. When all kinetochores are correctly attached to the microtubules, released APC/C-cdc20 ubiquitinates Cyclin B1 and thus induces anaphase onset [65].

Proper chromatid separation requires degradation of the link between sister chromatids, called cohesin [66]. During interphase, cohesin complexes are largely distributed to the whole chromosome, but most parts are removed during prophase mainly by the WAPL protein [66,67]. Only a small fraction of cohesin proteins remain on the centromere, which is protected by Sgo1 from WAPL proteins [66,68]. At anaphase onset, activated APC/C-cdc20 facilitates ubiquitination and degradation of securin, which suppresses the protease activity of separase [65], followed by cleavage of the cohesin complex by active separase [66].

### 3.2. The Role of ncRNAs in the Assembly and Function of Kinetochores

The centromeric DNA sequence where the kinetochore structure is formed does not encode proteins, but is transcribed by RNA polymerase 2 (RNA polII) in human cells [69,70]. RNA polII accumulates on the centromere from prometaphase to anaphase and disperses during interphase [69]. RNA polII at the centromere is phosphorylated at serine 2, which is a marker for active transcription [69]. Localization and transcription of RNA polII at the centromere requires activity of Bub1, a kinase related to the SAC [70]. The transcripts of the centromere are termed “centromeric RNAs”, which are capped in *Drosophila* [71]. Whereas centromeric RNAs in fission yeast are polyadenylated [72], those of human cells are not [73]. Several studies show that inhibited transcription or splicing in mitosis or degradation of centromeric RNAs results in reduced CENP-C intensity on the kinetochore, anaphase extension, and chromosome segregation errors [22,73,74,75]. However, a recent study suggested that transcription in mitosis was not required for mitotic progression in human cells [24]. The importance of transcripts in centromeric DNA has been reported for many species, including humans [75], maize [21,76], budding yeast [77], fission yeast [78], and flies [72].

The relationship between centromeric RNAs and kinetochore assembly was first investigated by experiments with maize in 2004 [21]. This study revealed that the centromeric retrotransposon and satellite-derived 40–200 nt transcripts bind to centromere histone H3. Since no other repetitive sequences or similar sequences were detected in the co-immunoprecipitated fraction against centromeric histones, it was concluded that centromeric RNAs are specifically associated with the kinetochore [21].

In 2007, experiments using human cells also revealed that transcripts of alpha-satellites in the centromere region bind to CENP-C. This interaction is necessary for accumulating CENP-C, INCENP, and survivin to the kinetochore. Furthermore, these three proteins were diminished from the kinetochore by RNase treatment; adding alpha satellite RNAs consistently recovered their accumulation at the kinetochore [22]. Unless otherwise stated, the following sentences primarily describe the functions of “centromeric RNAs” in humans [23].

In addition to CENP-C, centromeric RNAs also promote centromeric deposition of CENPA [75]. On the contrary, the centromere was transcribed even without CENP-A [75]. The length of the CENP-A-associated centromeric RNAs transcribed from alpha-satellite DNA ranged from 300 to 2000 nt in human cells [75]. The study also showed that the nascent centromeric RNAs stayed around the transcribed region of the centromeric DNA to participate in kinetochore function. Accordingly, depletion of chromosome-specific centromeric RNAs reduce the intensity of CENP-A and CENP-C only in the centromere where RNAs are transcribed. This system must be properly exerted for each chromosome since disrupting the kinetochore assembly results in cell cycle arrest in the S or G2 phase, even for a single chromosome [75].

The neocentromere is a DNA region where the kinetochore is formed only when the normal centromere is dysfunctional. Similar to a normal centromere, the L1 retrotransposon on the 10q25 neocentromere is transcribed in human cells [79]. The transcripts help incorporate CENP-A, which implies that transcription at the centromere, rather than its specific DNA sequence, is related to its function as a scaffold for kinetochore assembly [79]. Conversely, another study showed that the alpha-satellite sequence is actively transcribed even in a functionally inactive centromere. While transcripts from the active centromere bind to CENP-A and CENP-C and work in the kinetochore assembly, transcripts from the inactive centromere are unstable and bind to CENP-B [75].

A study using murine erythroleukemia cells reported that binding of chromosome passenger complex (CPC) to CENP-A and aurora-B kinase activity were attenuated by RNase treatment, while adding minor satellite RNA, which is a centromeric RNA, rescued this phenotype [80]. Furthermore, murine satellite RNA binds to CENP-A, as well as CPC components, aurora-B and survivin. Experiments using human cells and *Xenopus* egg extracts also showed that centromeric RNAs are associated with aurora-B and INCENP and promote their loading to the centromere [73,81]. Reduced centromeric RNA expression weakened the ability for aurora-B to monitor microtubule–kinetochore binding [81]. However, whether the kinase activity of aurora-B increased or decreased after removal of satellite RNAs is still controversial [73,81,82]. Furthermore, how the associated RNAs affects activity of Aurora-B remains unclear.

Sgo1 protects the cohesin complex at centromeres. Sgo1 recognizes Histone H2A-T120 phosphorylation by Bub1 [83] and thereby accumulates on the centromere, after which it migrates to the inner kinetochore. This transition from the centromere to the kinetochore is promoted by centromere transcription by RNA polII. Inhibiting transcription during mitosis removes Sgo1 from the inner kinetochore and thus weakens centromere cohesion [70]. Although Sgo1 binds to alpha-satellite RNAs, whether the transcripts or transcription itself is significant for Sgo1 function remains unclear.

Overall, the function and significance of centromeric RNAs during assembling of major kinetochore and centromere proteins, such as CENP-A, CENP-C, aurora-B, and Sgo1 were introduced (Figure 3). Since centromeric RNAs are transcribed and accumulated during mitosis [69,80], inhibiting transcription in mitosis affects kinetochore function [69,70]. However, a recent study suggested that transcription in mitosis was not required for mitotic progression [24]. The authors reported that DNA-intercalating drugs (α-amanitin) reduce localization of aurora-B in the centromere without reducing the amount of centromeric RNAs [24]. Therefore, they claimed that structural change of centromeres affected aurora-B loading. Future studies will distinguish the role of centromeric DNA transcription, transcripts, and structure of centromeric DNAs in kinetochore assembly.

Although this review mainly focused on the role of centromeric RNAs on human kinetochore assembly, but the significance of centromeric transcripts are evident in animals, plants, and fungi. Since the sequence of centromeric DNA is little conserved among species, physiochemical properties, such as single-strand negatively charged RNA molecules, may be more important than the sequence itself, such as in the formation of open chromatin structures [84]. Interestingly, in experiments with maize, any single-strand RNA and DNA with long nucleotides, including centromeric RNAs, were attached to the DNA-binding domain of CENP-C to promote DNA-binding activity and centromeric accumulation [76]. In addition, aurora-B kinase binds to any RNA with adenine-rich sequences to increase its activity [82]. Studies on the neocentromere also suggested the importance of RNA transcription within the region rather than its sequence [75]. Therefore, the important properties of centromeric RNA may be that it is a long, single-strand structure, and, most importantly, generated and remains in the centromere.

## 4. Mitotic Spindle and Microtubules

### 4.1. Microtubule Dynamics and Mitotic Spindle Formation

Eukaryotic cells have three types of cytoskeletons—microtubules, actin fibers, and intermediate filaments. Here, we focus on microtubules, which contribute to cell morphology maintenance, intracellular transport, cilia formation, and cell division. Microtubules possess a hollow structure with 25 nm diameter made up of thirteen identical polymers (protofilament) of the α-βtubulin heterodimer. Tubulin polymers are highly dynamic and frequently grow and shrink, a characteristic called “dynamic instability”. Microtubule stability is regulated by post-transcription modification (PTM) and microtubule-associated proteins (MAPs) [85,86,87]. Among the MAP proteins, those that bind to the microtubule plus ends are called “+TIPs”; they include XMAP215, a microtubule stabilizer, and MCAK, a microtubule destabilizer [88], among others.

In mitosis and meiosis, microtubules form a rugby-ball-like mitotic/meiotic spindle critical for properly distributing sister chromatids to daughter cells. Spindle microtubules mainly nucleate at the centrosome, but chromosome-dependent and augmin-dependent pathways also exist [89]. The chromosome-dependent pathway involves importin, Ran GTPase, and the chromosome passenger complex [90,91]. On the other hand, the augmin complex is present on microtubules and induces branching microtubule nucleation [89,92].

### 4.2. The Role of RNAs and RNA-Binding Proteins in Mitotic Spindles

In this chapter, the role of RNA and RNA-binding proteins in microtubules and mitotic spindles are introduced (Figure 4). Given that treating human cells with transcription inhibitors [74] or RNase, but not translation inhibitors [17], destabilizes the mitotic spindle structure, it can be assumed that functional ncRNAs or mRNAs [40,41] act as regulators or structural components of the mitotic spindle. One example is the regulation of aurora-kinase-B and MCAK. MCAK is a microtubule depolymerizer whose activity is suppressed by aurora-kinase B. Since the association of aurora-kinase B with RNAs is somehow required for proper localization and function of aurora-kinase B, inhibiting RNA transcription or RNase treatment in *Xenopus* egg extract attenuated kinase activity and thereby increased MCAK density on mitotic spindles, which lead to microtubule depolymerization. In contrast, transcription inhibitors or RNase treatment reduced the density of TPX2, a spindle assembly factor implicated in chromosome-dependent MTOC activity on mitotic spindles. Interestingly, inhibiting MCAK or adding purified TPX2 proteins relieved spindle destabilization caused by treatment with transcription inhibitor or RNase. These studies imply a non-coding function of RNA molecules as an activator for their interacting partner proteins [60].

In 2007, Blower et al. performed a microarray analysis on purified mitotic spindles from *Xenopus* egg extract and identified enriched mRNAs and rRNAs [93]. The enriched mRNAs labeled with fluorescence were detected on the mitotic spindle by microscopy [93]. Although mRNAs concentrated on the mitotic spindle were translated, the translation itself was not necessary for their localization. Furthermore, mRNAs with a cytosolic polyadenylation element (CPE) in 3′UTR tended to accumulate in the mitotic spindle fraction. The authors also applied the method to purified mitotic spindles of HeLa cells and found a similar tendency. In 2009, another study using a DNA probe showed a specific localization of 18S rRNA on the mitotic spindle [17]. 18S rRNA-binding protein Misu was also translocated to the mitotic spindle from the nucleoli, and this migration was attenuated by RNase treatment. Misu is an essential protein for bipolar spindle formation since its depletion results in multipolar spindle formation. Although the major function of Misu is thought to be RNA methylation downstream of c-Myc signals [94], the methyltransferase activity of Misu is not required for normal mitotic spindle formation; this suggests that only RNA-binding activity is important for proper bipolar spindle formation. Therefore, ncRNAs on the mitotic spindle can partake in spindle formation by recruiting RNA binding proteins.

The microtubule-binding protein Rae1 is also essential for bipolar spindle formation in HeLa cells [16]. Rae1 is an importin β-binding protein responsible for mRNA export in ribonucleoproteins (RNP). In *Xenopus* egg extract, Rae1 is necessary for activating Ran-GTP to form microtubule asters, as well as for spindle assembly following sperm nuclei injection. RNase treatment can reduce aster nucleation from Rae1-beads in *Xenopus* egg extract [16], demonstrating that RNAs are critical for Ran-GTP-dependent MTOC activity.

In addition to the aforementioned proteins, several RNA-binding proteins, such as spermatid perinuclear RNA-binding protein (Strbp) [95,96], Staufen [97], and adenomatous polyposis coli (APC) [98] directly interact with microtubules. Although these proteins contribute to mRNA transport in microtubules, they can also bind to ncRNAs or mRNAs that directly regulate microtubules. Future studies will identify further binding RNAs and reveal their function in microtubule regulation and mitotic spindle formation.

In addition, ncRNAs can function as a seed for spindle matrix assembly through phase-separation. A spindle matrix is a protein structure observed around mitotic spindles that promotes spindle assembly [99]. Since spindle matrixes maintain their localization after depolymerization of spindle microtubules by nocodazole treatment, they are considered an elastic matrix [100]. Purified BuGZ, a major spindle matrix component, forms phase-separated condensates in vitro [101]. However, whether BuGZ shows liquid-like behavior in vivo has not been elucidated [50].

## 5. Concluding Remarks

Pericentriolar material (PCM) is a mass of proteins surrounding centrioles. Considering that many proteins with an intrinsic disordered region exist in PCM, PCM may be formed by phase separation of its components. In 2017, a study using nematodes showed that the PCM protein SPD-5 has a property of phase separation in vitro and in vivo, and that the SPD-5 condensates promote microtubule nucleation in vitro when mixed with tubulin, microtubule polymerase ZYG-9 and microtubule-stabilizing protein TPXL-1 [50]. RNAs may serve as a seed for the phase separation of PCM. Observations using super-resolution microscopy revealed that PCM has a layered structure [102]. As NEAT1 forms a layered structure of nuclear paraspeckles by binding to proteins in a sequence-specific manner, there may be an RNA molecule which similarly contributes to the layering of PCM. Such RNA has not been identified currently, but may be determined in future studies.

As for mitotic spindle formation, a large number of RNA-binding proteins are present on mitotic spindles and have been shown to be required for their formation. Although the reported binding partners of these RNA-binding proteins are mostly mRNA, it is possible that these RNA-binding proteins also bind to a specific ncRNA which structurally functions in mitotic spindle assembly. The function of ncRNA may be to induce phase separation of proteins such as BuGZ, or to assemble several components or regulators that cooperatively promote mitotic spindle assembly.

Several groups previously performed a knockdown screening using RNAi [103] or CRISPRi system [104] and identified several lncRNAs involved in cell division [103]. Some of these lncRNAs may directly function for mitotic apparatus formation. Although most of the reported functional RNAs are related to the regulation of gene expression, future approaches will reveal functional RNAs structurally involved in mitotic apparatus formation.

## Figures and Tables

**Figure 1 ncrna-06-00013-f001:**
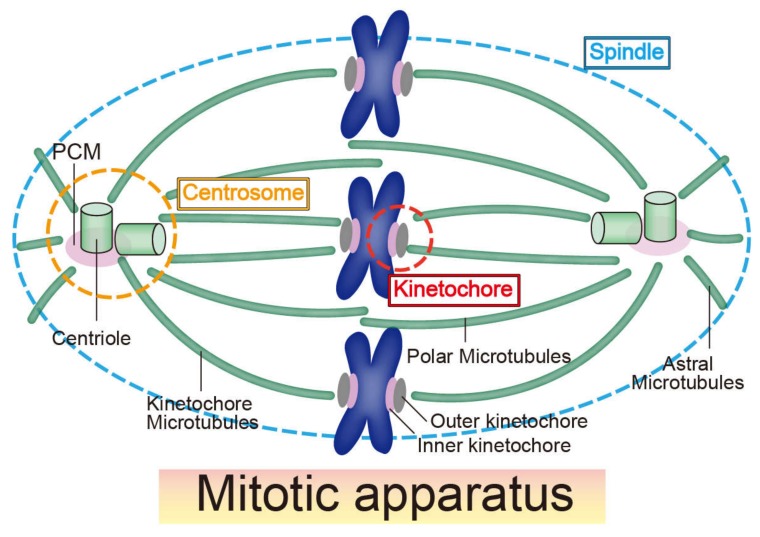
Schematic illustration of the mitotic apparatus. The mitotic apparatus is composed of centrosomes, kinetochores, and a mitotic spindle. All three structures comprising the mitotic apparatus accumulate functional RNAs and RNA-binding proteins.

**Figure 2 ncrna-06-00013-f002:**
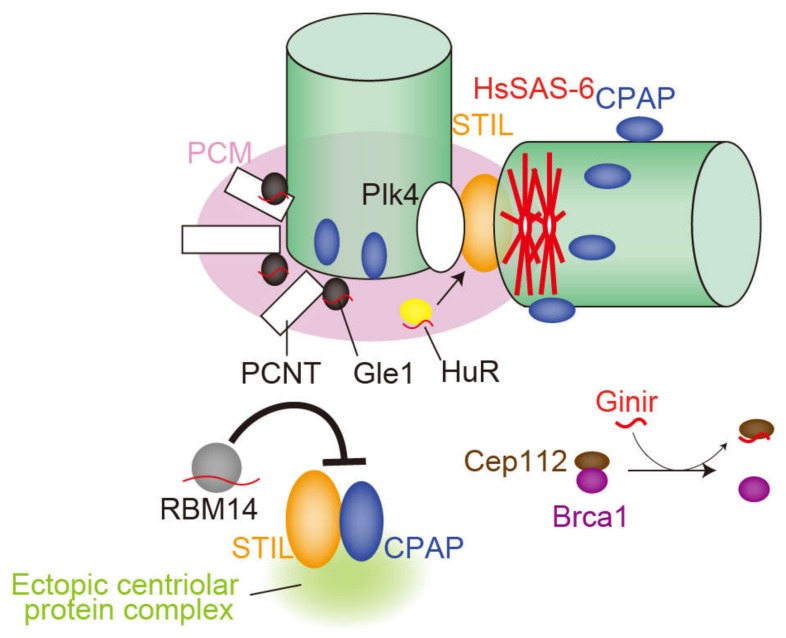
The role of RNAs and RNA-binding proteins (RBPs) in mammalian centrosomes, described in Chapter 2. The red curve denotes RNA. Ginir is a murine long non-coding RNA (lncRNA). RNAs associated with Gle1, HuR, and RBM14 may be messenger RNA (mRNA) or non-coding RNA (ncRNA).

**Figure 3 ncrna-06-00013-f003:**
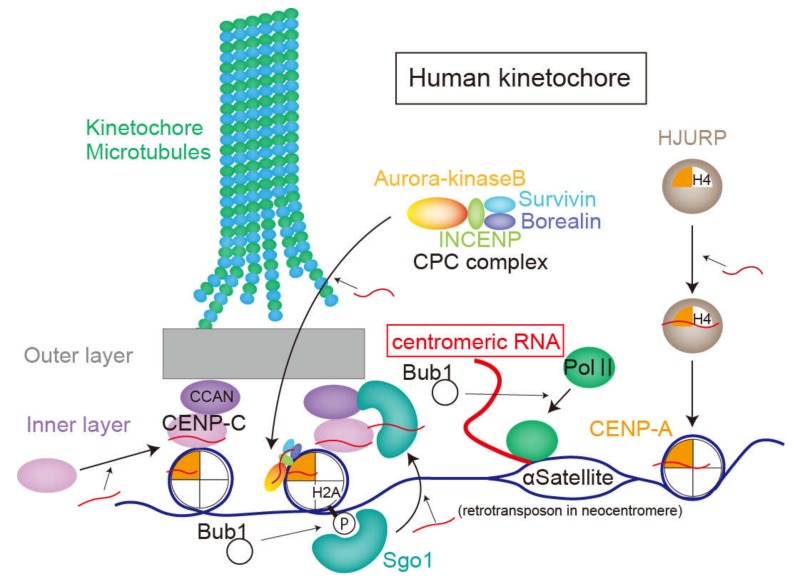
The role of centromeric RNAs in human kinetochores, described in chapter 3. Red curves show centromeric RNAs transcribed by polII. The transcripts recruit CENP-C, CENP-A, Sgo1, and chromosome passenger complex (CPC), including aurora-kinase B to the centromere or kinetochore.

**Figure 4 ncrna-06-00013-f004:**
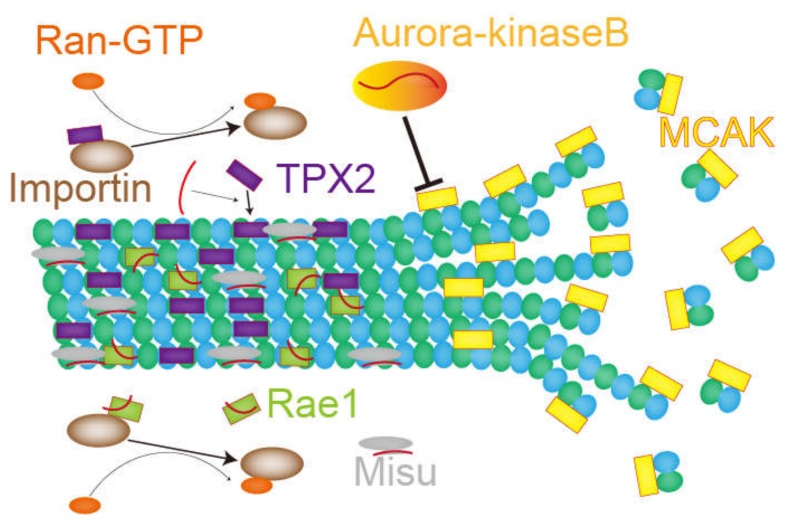
The role of ncRNAs on microtubules, described in chapter 4. Red curves show RNAs. The RNA associated with Misu is 18S rRNA. Other RNAs are ncRNAs or perhaps mRNAs necessary for nucleation and stability of microtubules.

**Table 1 ncrna-06-00013-t001:** List of RNAs and RBPs in the centrosome introduced in this review.

RNAs and RBPs in the Centrosome	Function
RNAs	
cnRNAs	Unknown
Ginir	To perturb the binding between Cep112 and Brca1
RNA binding proteins	
RBM14	To perturb the binding between STIL and CPAP
Gle1	Maintenance of the level of PCNT in PCM
HuR	The regulation of centrosome number
RBM8A and MAGOH	The regulation of centrosome number
EWS/NDH2	Unknown

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
