# Peer review of "The Emerging Role of ncRNAs and RNA-Binding Proteins in Mitotic Apparatus Formation"

_ncrna, 2020, doi:10.3390/ncrna6010013_

Round 1
Reviewer 1 Report
The manuscript entitled " The role of ncRNAs in mitotic apparatus formation" by Ito et al. aims to review the accumulating evidence for a critically important role of non-coding RNAs in organizing the mitotic apparatus.
However, this manuscript unsuccessfully balances classic textbook knowledge with recent findings in the field. Importantly, the text does not move ncRNAs and their biology into the focus of the potentially interested reader. Specifically, the manuscript uses up most of the available space by giving three times a rather general introduction into the biology of the core components of the mitotic apparatus (often without even mentioning the term RNA). And even when the authors aim at the supposed focus of this review (ncRNA), they often intersect the mentioning of other RNA species than ncRNA (i.e., mRNA) with the biology of various RNA-binding proteins which have been characterized through their interactions with mRNAs and not ncRNAs.
While the multitude of reviews is written in this fashion, this reviewer strongly feels that a good review should abstain from just re-iterating the facts but should aim at generalizing findings, synthesizing commonly applicable conclusions, and at making the readers become interested in open questions to be solved.
It follows that in the present form, this review is not contributing much to the general public record and very little to the ncRNA field and, therefore, should not be considered for publication in a journal with the name “Non-Coding RNA”. Hence, the authors should restructure and focus the text on RNA before the manuscript can be re-considered for publication.
General Comments:
(1) Re-structuring of the text:
Combine and cut short the general comments about the components of the mitotic apparatus.
Introduce the history of detecting RNA in the mitotic apparatus first for readers to understand where the problems of studying ncRNAs in this context lie(d). For instance, paragraph 2.2 and the history of detecting RNA in centrosomes could serve as the basis for a concise description of what is known about the RNA components in the mitotic apparatus.
(2) Focus on ncRNAs at the expanse of describing the function of mRNA-binding proteins:
The authors might want to convey the picture that all kinds of RNAs have been localized to mitotic apparatus (i.e., page 3: 120 cnRNAs) but while the function of mRNAs has been solved (is that so?), the function of ncRNAs remains unclear. Then home in on what is known about the biology of ncRNAs and why ncRNAs would be suitable for organizing the mitotic apparatus.
(3) Are there species-specific ncRNAs?
Separate the identification of ncRNAs between species in order to make the reader understand whether such ncRNAs might be conserved. Presently, the text jumps between all kinds of findings in different organisms, which is confusing. This has been addressed indirectly by discussing the sequence-independence of “centromeric” RNAs but should be expanded.
(4) Explore the functional/scaffolding/solubilizing role of ncRNAs:
It is unclear why the authors introduce LLPS and the structural roles of RNAs in the formation of membrane-less organelles late and only in the discussion. Given that ncRNAs are often not conserved, it follows that it is physicochemical properties of the polynucleotide RNA that are underlying RNA function in processes where aggregation of ribonucleoprotein particles is required for cellular function. The process of transcription of centromeric RNAs is mentioned in the text and generalizing such observations might be provocative and meaningful to the reader. In this respect, a recent paper by the Farnebo-group (PMID: 31447351) reported on single-stranded RNA to be important for affecting chromatin condensation in general, which might be of importance for understanding why abundant RNAs (such as transposon-derived RNAs) are often expressed in cells.
Reviewer 2 Report
Two major concerns: a) restructuring the text will help because authors quickly jump from describing one interaction to another in the text. Rather they should be focus on specific interactions. b) Some of the figures are difficult to understand without proper description in the legend or caption.Author Response
Please see the attachment.

Reviewer 3 Report
In this manuscript, Ito et al aim at reviewing the recent literature regarding the possible regulatory roles of (nc)RNAs and RNA-binding proteins (RBPs) in the formation & function of the mitotic apparatus.
The mitotic apparatus is composed of three main structures: centrosomes, kinetochores and mitotic spindles. After a short general introduction, each of these structures is described in a specific section, each subdivided into two subsections, including (i) a general description of the structure of interest, and (ii) a paragraph focusing on the roles of (nc)RNAs and RBPs in this structure. I like this organization of the manuscript.
However, one could regret the review does not provide clear mechanistic insights into the direct and precise function(s) of ncRNAs within the mitotic apparatus, but this is probably because this remains largely unknown to date (lines 397-8). The review gives a large place to RBPs and their roles in the mitotic apparatus, so that this could appear in the title (see minor point 1).
Here are some recommendations that could improve the quality of the manuscript.
Major point:
Figures 2-4 should be improved. These figures are really difficult to interpret and understand, especially due to the lack of caption (at least, the color code should be clarified and a key should be provided). In addition, the main text does not refer to any of them, so it is quite unclear to understand what these figures aim at illustrating. For example, in Fig 3: in the main text, it is written that centromeric RNAs promote the loading of Aurora-B and INCENP proteins to the centromeres. This does not appear at all in the figure. Furthermore, it is said that Bub1 phoshorylates histone H2A-T120, but why the arrow from Bub1 to the phosphate group of RNA PolII in the figure?
Minor points:
I would recommend to modify the title into: ”Roles of ncRNAs and RNA-binding proteins in mitotic apparatus formation”. Line 23-24: the authors could provide more accurate numbers of transcripts. For example, the current version of LNCIpedia (5) includes 127802 transcripts. More recent references could be given. Lines 36 and 39. Please mention the exact references at the end of the sentences. Lines 106-9. Please provide a more accurate description of Ginir. It is first presented as “a mammalian lncRNA”, then as a sense/antisense transcript pair. There are indeed two distinct transcripts, Ginir (sense) and Giniras (antisense). Please clarify the roles of each transcript. Line 117. Provide a reference or a link for the MicroKits database. Section 3.1 (lines 162-213). It is not clear to me if the described process is conserved across all Eukaryotes and whether all the cited studies are related to human cells. If necessary, please clarify the model organism in which the cited data have been obtained. Line 247-8. To which study do the authors refer when writing “… reported to be 1300 nt in the study”? Would it be possible to describe more into the details the features of CENP-A-associated centromeric RNAs? Lines 321 and 331. The authors point a possible role of “ncRNAs”. However, one could imagine mRNAs with dual functions, as previously proposed (for example, see PMID: 21111023) Check the format of the references list. For example, ref. 35 is not correctly formatted (the name of the journal is missing). Table 1 is not mentioned in the main text.Author Response
Please see the attachment.

Reviewer 4 Report
Summary
Work over the past decade has indicated that mRNAs and ncRNAs localize to various parts of the mitotic apparatus where they contribute to successful mitotic division. This review summarizes the RNAs that localize to centrosomes, kinetochores, and the mitotic spindle. This review does a nice job of summarizing the contributions of ncRNA to mitosis and will be a valuable addition to the literature. I have a few minor suggestions that I think could improve accuracy and readability.
In section 1 the authors discuss the contributions of ncRNA to the centrosome. At the end of the review they discuss recent literature showing that the centrosome is a phase separated organelle. Since phase separation is so closely associated with RNA and RNA-binding proteins I think it would make more sense to include the phase separation discussion in section 1 of the review. Line 117. Please explain MicroKits database. Throughout the review the authors point out that mutations in mRNA splicing factors lead to mitotic defects. The authors should point out the possibility that these mitotic defects could be caused by errors in mRNA splicing, rather than a direct role in mitosis. Line 197. CPC is localized to the inner centromere. Line 216. References 67, 68 are incorrect. Line 221. Inoue et al. 2014. Find that alpha satellite RNA is not polyadenylated. This should be mentioned. Line 223. A recent study that inhibited transcription specifically during mitosis found that there are no mitotic defects (Novais-Cruz et al. eLife 2108). This should be mentioned. Line 242-243. I would be careful including reference 72. There has been a significant correction issued by these authors and I am not sure that much weight should be attributed to the results published in this study. Line 350. Blower et al. 2005 did not show that Rae1 is sufficient for Ran-GTP induced spindle assembly.
Round 2
Reviewer 1 Report
This reviewer thinks that the review has improved considerably, but suggests to address some additional shortcomings and comments before the review can be published in “Non-Coding RNA”.
General Comments:
1) >> Introduction of chapters by using the term: “General statement” seems odd and like an administrative warning rather than the necessary introduction to describing the details of a biological system. The authors should use a chapter header that conveys specificity for the content that will be discussed.
2) >> Still, these “General statements” do not alert the reader on what is coming next in terms of ncRNA. Clearly, these “statements” could be cut even further without loss of information. Since there is much more known about each of these subcellular structures, it seems odd to add some detail and then cut off without transition to the next chapter addressing ncRNAs. The authors could mention what is not known about centrosomes, kinetochores, the mitotic spindle and microtubules with a hint towards lack of knowledge regarding if RNA and which RNAs contribute to the structure and function of these components.
3) >> The authors still did not introduce LLPS and the structural roles of RNAs in the formation of membrane-less organelles but have moved the mentioning from the discussion to chapters 2 and 4. This reviewer repeats a previously made statement: given that ncRNAs are (often) not conserved, it is likely that the physicochemical properties of ncRNAs are underlying its function, especially in processes when aggregation of ribonucleoprotein particles is required for cellular function such as the formation of subcellular structures. Therefore, the authors should introduce some knowledge about ncRNAs in the formation of such structures into the introduction, preferably around this sentence: “Some lncRNAs function as scaffolds during the construction of subcellular structures, such as nuclear bodies [6–9].” Here, the authors could also mention particular RNAs contributing to LLPS such as NEAT to make the reader aware of the possibilities that other ncRNAs could contribute to assembling the components of the mitotic spindle apparatus.
4) >> The “Concluding remarks section” is rather short and not really satisfying to the reader. “In this review, we discussed the role of ncRNAs and RNA-binding proteins during the formation of the mitotic apparatus, which is composed of centrosomes, kinetochores, and spindle microtubules.” is a redundant statement because the review is all about it and that does not need to be mentioned again.
It would be helpful for the impact on the reader not to repeat what has been already mentioned in the main text but to identify OPEN questions in the field.
- Which RNAs contribute to mitotic spindle apparatus function?
- Sequence context or physicochemical properties of an RNA that contribute to mitotic spindle apparatus function?
- Do we need structural knowledge about ncRNAs in order to arrive at answers?
- Do RNA modifications play a role in the function of ncRNAs contributing to mitotic spindle apparatus function?
5) >> Replace an often used phrase in the manuscript “In this chapter, we…” with some non-personal statements.
Specific Comments:
PAGE 4: “ncRNAs may be involved in PCM nucleation by acting as a seed for LLPS.”
>> The abbreviation LLPS has not been defined. Also, the authors use different terms for similar identifiers such as “phase-separated condensate”, “liquid-phase condensation” but not liquid-liquid phase separation. This should be unified throughout the text.
Reviewer 2 Report
I do not have any major concerns on the revised version of the manuscript, except it felt like that in some instances there could be better choice of words and phrases.
